# A new potential strategy for cutaneous squamous cell carcinoma treatment by generating serum-based antibodies from tumor-exposed mice

Zheng Liu*

College of Medical Laboratory Science, Guilin Medical University, Guangxi, China

**Abstract** Current cancer treatment strategies continue to face significant challenges, primarily due to tumor relapse, drug resistance, and low treatment efficiency. These issues arise because certain tumor cells adapt to the host immune microenvironment and evade the immune system. This study presents a new cancer immunotherapy strategy using serum-based antibodies from mice exposed to mouse cutaneous squamous cell carcinoma (mCSCC). The experiment was conducted in three stages. In the first stage, mCSCC cells were isolated and expanded cultured from DMBA/TPA-induced mCSCC. In the second stage, these expanded tumor cells were injected into healthy mice to stimulate the production of anti-tumor antibodies. In the final stage, therapeutic serum was extracted from these healthy mice and reintroduced into the tumor-bearing mice. An ELISA assay was utilized to analyze the levels of p53, Bcl-xL, NF-κB, and Bax. The results showed that the serum treatment not only reduced tumor volume but also reversed changes in p53, Bcl-xL, NF-κB, and Bax. In conclusion, this study developed a new immunotherapeutic strategy for treating mCSCC. However, further research is needed to fully comprehend the mechanism of this serum treatment.

*For correspondence:
zliu1111@163.com

Competing interest: The author declares that no competing interests exist.

## eLife assessment

This study provides a **valuable** strategy for treating mouse cutaneous squamous cell carcinoma (mCSCC) with serum derived from mCSCC-exposed mice. The exploration of serum-derived antibodies as a potential therapy for curing cancer is particularly promising but the study provides **incomplete** evidence for specific effects of mCSCC-binding serum antibodies. This study will be of interest to scientists seeking a novel immunotherapeutic strategy in cancer therapy.

## Introduction

The primary challenges in cancer treatment today include cancer heterogeneity, therapeutic resistance, and tumor recurrence (*Dagogo-Jack and Shaw, 2018*). The predominant strategies for cancer therapy currently encompass chemotherapy, radiotherapy, and immunotherapy. Despite significant strides made in cancer treatment over the past decades, issues persist with resistance to traditional chemotherapeutic agents and a lack of specificity in targeting cells (*Wang et al., 2019*). Certain cell surface proteins have emerged as valuable targets and biomarkers for cancer therapies. However, high tumor recurrence rates remain a significant concern. This is primarily due to the tumor cells expressing different biomarkers at different developmental stages (*Poudineh et al., 2018*). Within a tumor mass, there coexists a multitude of tumor cells at different stages and of diverse types. Some of these cells can evade treatment targets when subjected to chemotherapy and radiotherapy that only target one or several biomarkers (mutated proteins) (*Garg et al., 2016*). These evasive tumor cells, particularly

**Figure 1.** The workflow of this study.

the cancer stem cells with self-renewal and differentiation capabilities, undergo genetic alterations and modify cell-surface antigen production (mutated proteins) to evade the immune system (*O'Donnell et al., 2019*). Unexpectedly, these strategies may inadvertently create a conducive growth environment for these evaded tumor cells.

Immunotherapy is designed to strengthen the patient's immune system in order to eradicate tumor cells (*Barrett and Puré, 2020*). There are currently several types of immunotherapy utilized in cancer treatment, which include immune checkpoint inhibitors, T-cell transfer therapy, and monoclonal antibodies (*Marin-Acevedo et al., 2018*). Despite significant improvements in both active and passive cancer immunotherapy over recent years, these methods have not completely succeeded in preventing the recurrence of tumors (*Jackson et al., 2019*). The primary reason is the ability of some tumor cells to adapt to the immune microenvironment and evade the immune system by altering the expression or structure of proteins, preventing immune cells from recognizing them as foreign antigens (*Beatty and Gladney, 2015*). Furthermore, the aforementioned methods do not consider the tumor mass as a whole entity, which encompasses cancer cells at various developmental stages, each harboring a range of known and unknown mutated proteins. This implies that the pattern of tumor markers (mutated protein) associated with each individual's tumor is unique. As a result, these therapeutic approaches are unable to completely eradicate tumor cells across diverse types and stages. In this study, a new cancer treatment strategy is designed using mCSCC as a model. This strategy aims to treat tumors in three stages: isolating tumor cells, producing serum-based antibodies, and eliminating the tumor cells.

## Results

### Monitoring body weight

Fifty male C57BL/6 mice were randomly divided into five equal groups: tumor + serum treatment, tumor + no serum treatment, control + serum treatment (control 1), control + no serum treatment (control 2), and serum provider (*Figure 1*). At the beginning of the experiment, male C57BL/6 mice (aged 6–8 weeks) had an average body weight of 20.5±0.3 g, with a range of 20.0–21.3 g. Following 12 weeks of DMBA/TPA treatment, mCSCC developed on the backs of these mice in tumor + serum treatment and tumor + no serum treatment groups. The average body weights of the DMBA/

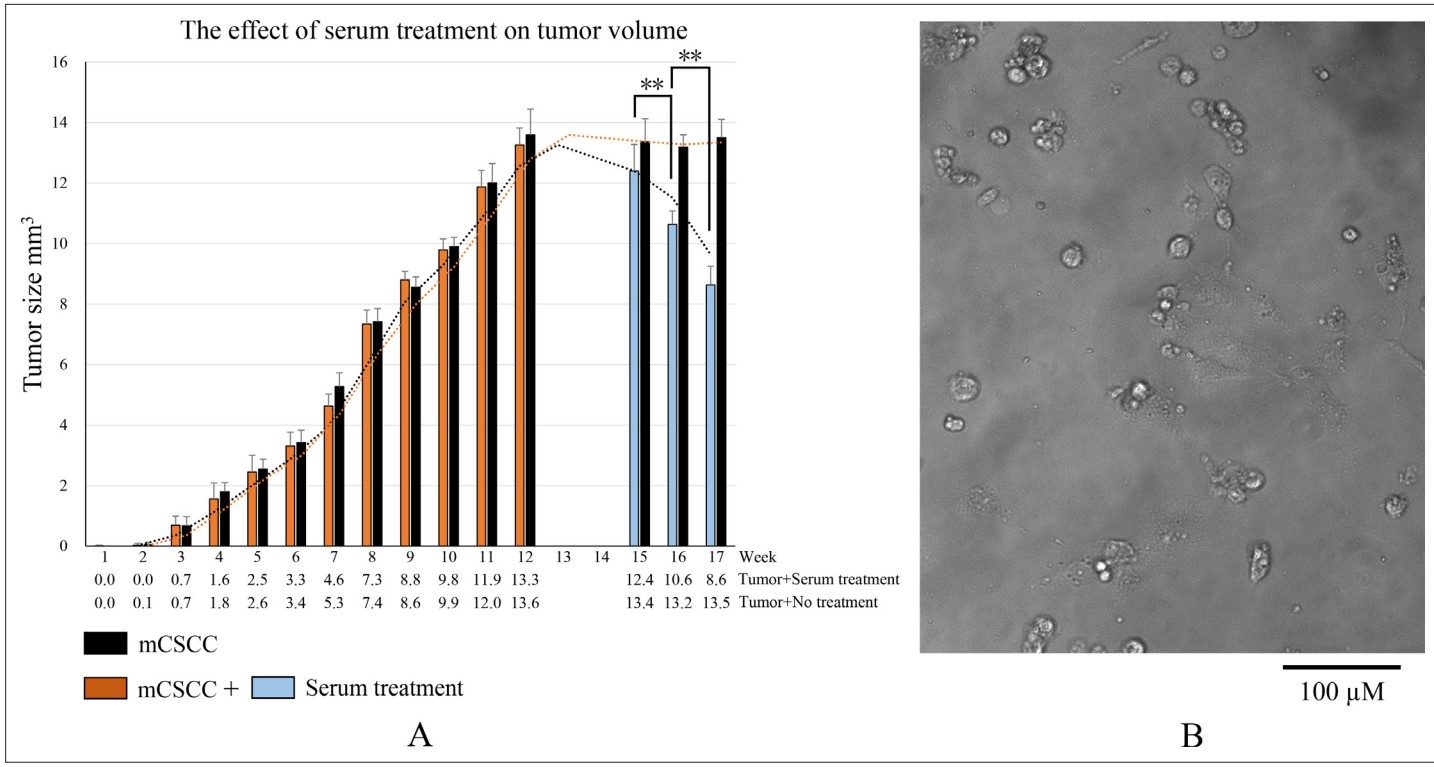

**Figure 2.** The serum-based antibodies treatment reduces tumor volume. (**A**) Tumor growth induced by Dimethylbenz(a)anthracene (DMBA)/Tetradecanoylphorbol-13-acetate (TPA) and changes in tumor volume before and after serum treatment. At week 12, the tumor volume reached its peak. Week 13 was dedicated to the isolation and expansion of the tumor cell. In week 14, the tumor cells were injected into the tail vein of paired mice in the serum provider group to produce serum-based antibodies. Weeks 15, 16, and 17 represent mice in tumor + serum treatment group receiving three times of serum treatment, respectively. Results are presented as the mean ± standard deviation (SD) obtained from at least three biological replicates. A paired two-tailed *t*-test was used for statistical evaluation. Asterisks indicate the following: **p<0.01. (**B**) Tumor cells isolated and cultured from mouse cutaneous squamous cell carcinoma (mCSCC). Scale bar, 100 μm.

The online version of this article includes the following source data for figure 2:

**Source data 1.** Excel file of mouse weight and tumor volume data is shown in *Figure 2*.

TPA-treated and control animals were 24.9±1.1 g and 26.7±0.8 g, respectively. At the end of the experiment (week 17), the average body weights were as follows: 26.6±1.4 g for the tumor + serum treatment group, 27.6±1.2 g for the tumor + no serum treatment group, 28.5±0.8 g for the control 1 group (control + serum treatment), and 28.5±0.8 g for control 2 group (control + no serum treatment).

## Serum treatment inhibits the growth of mCSCC

During the DMBA/TPA induction phase, the tumor progressively grows, reaching its peak average volume at 12 weeks. This volume measures 13.3 mm³ in the tumor + serum treatment group and 13.6 mm³ in the tumor + no serum treatment group. In the group that did not receive serum treatment, no significant changes in the tumor volume (13.5 mm³) were observed by week 17. However, after 3 weeks of serum treatment, the tumor volume dramatically reduced to 8.6 mm³ in the tumor + serum treatment group. This substantial decrease demonstrates the efficacy of serum treatment in reducing tumor volume (*Figure 2A*).

## Serum treatment reverses the expression of cancer biomarkers

The ELISA assay results indicate that in mCSCC, the expression levels of p53, Bcl-xL, and NF-κB are high, while Bax is expressed at a lower level. However, following serum treatment, the levels of p53, Bcl-xL, and NF-κB decreased, whereas the expression of Bax increased (*Figure 3*). These findings suggest that serum treatment can effectively reverse the expression of cancer biomarkers.

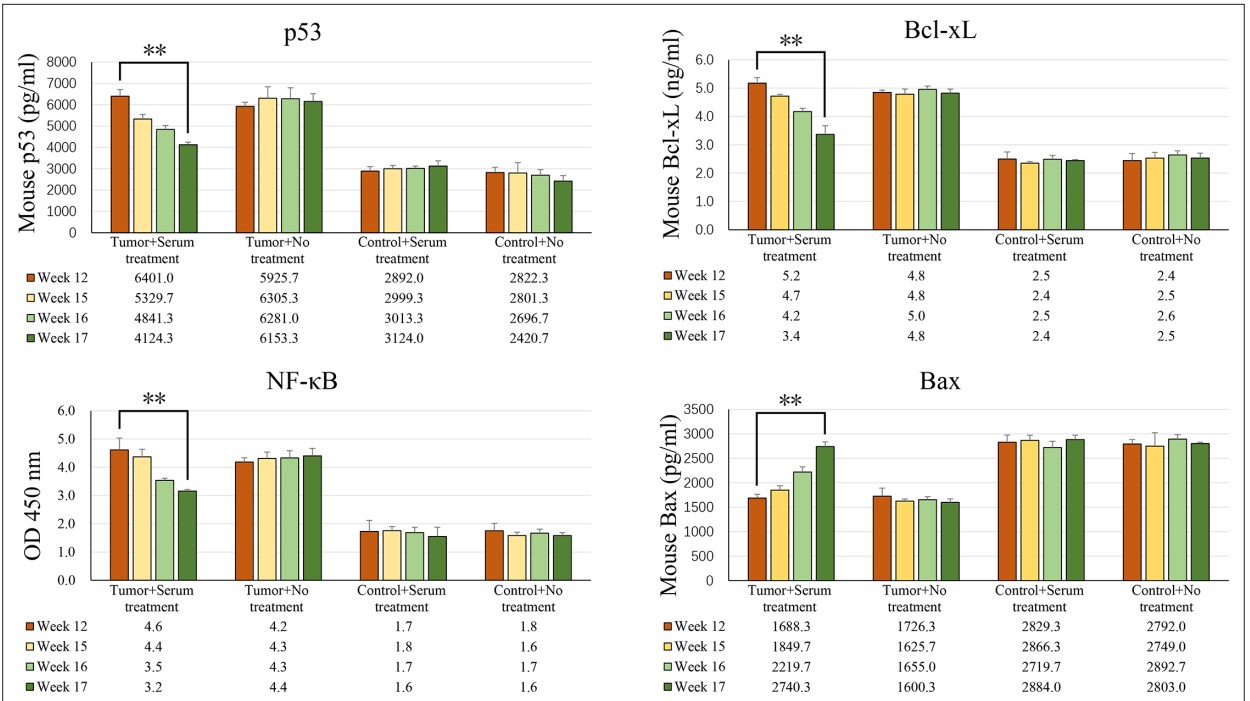

**Figure 3.** The serum-based antibodies treatment reverses the expression of p53, Bcl-xL, NF-κB, and Bax. ELISA analysis revealed the changes in the expression of p53, Bcl-xL, NF-κB, and Bax proteins before and after serum treatment. The tumor volume reached its peak at week 12. The mice in the tumor + serum treatment group received serum treatment at weeks 15, 16, and 17, respectively. Results are presented as the mean ± standard deviation (SD) obtained from at least three biological replicates. Statistical significance was determined by a paired two-tailed *t*-test. Asterisks indicate the following: **p<0.01.

The online version of this article includes the following source data for figure 3:

**Source data 1.** Excel file of ELISA analysis data for p53, Bcl-xL, NF-κB, and Bax expression is shown in **Figure 3**.

## Discussion

The principle behind developing this immunotherapeutic strategy is to treat various stages and types of tumor cells in the tumor mass as a whole entity. The various mutated proteins on tumor cells would be sensitively recognized as foreign objects and generate corresponding antibodies in a healthy individual (*Kallingal et al., 2023*). Although some tumor cells may evade the patient's immune system, they still can stimulate the production of serum-based antibodies in healthy mice. This strategy is divided into three stages: isolating tumor cells, producing serum-based antibodies, and eliminating the tumor cells (reducing tumor volume). After isolating the tumor cells, the cells of various growth stages and types in a culture medium were expanded. Injecting these cells into healthy mice led to the production of thousands of antibodies against the corresponding antigens (mutated proteins) on the tumor cells. The serum from the blood of these healthy mice was then transfused back into the tumor-bearing mice to treat mCSCC. Given that different stages of tumor cells have distinct surface biomarkers (*Woodward and Sulman, 2008*), the serum treatment procedure were repeated weekly for a total of three times (from week 15–17). The findings revealed a significant reduction in the tumor volume of the mice. *Figure 4* illustrates the principle and process of this experiment. To validate this treatment strategy, p53, Bcl-xL, NF-κB, and Bax, four mCSCC-associated proteins were selected as tumor biomarkers. In mCSCC, there was a notable increase in p53, Bcl-xL, and NF-κB, and a decrease in Bax. Serum antibodies for p53, Bcl-xL, NF-κB, and Bax were produced after injecting tumor cells into healthy mice. The tumor volume decreased following the serum treatment, which was accompanied by a reversed change in p53, Bcl-xL, NF-κB, and Bax levels. Regrettably, one healthy mouse from the serum provider group and one tumor mouse that received serum treatment died during the study due to unknown reasons.

As early as 1973, research demonstrated that the transfer of serum antibodies could decelerate tumor growth (*Anonymous, 1973*). However, following this discovery, research on serum therapy for

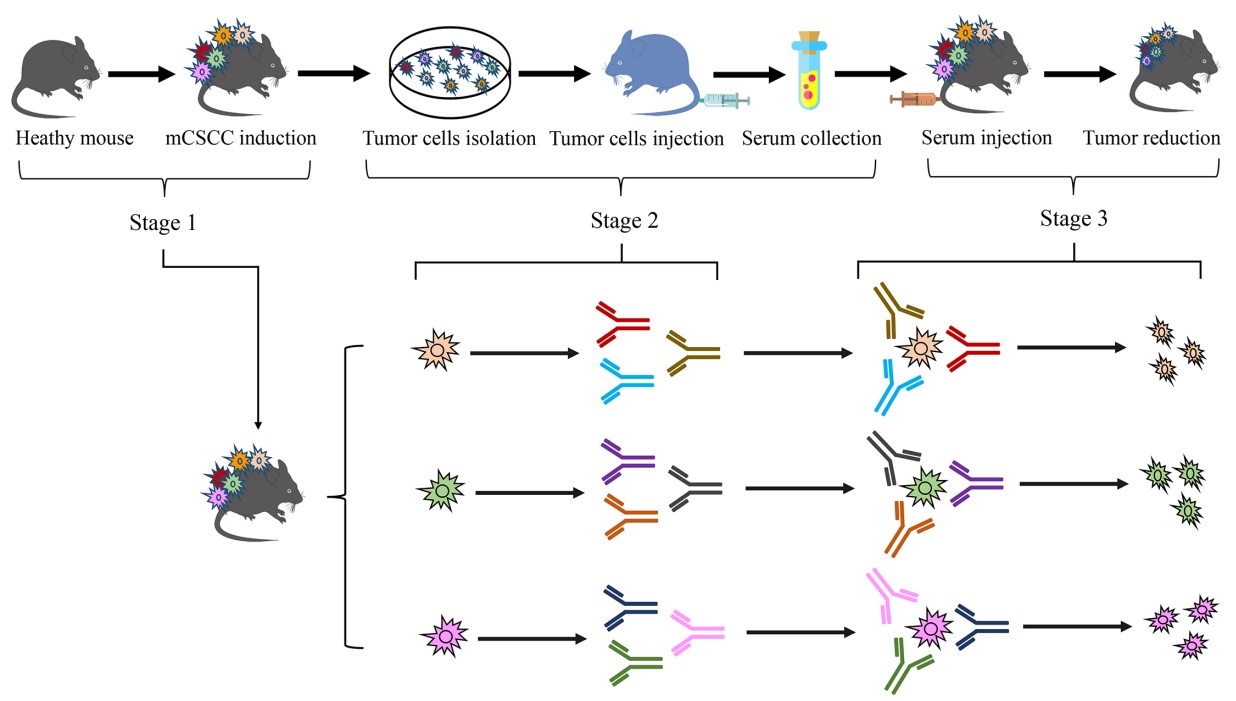

**Figure 4.** Schematic diagram of experimental design.

tumors nearly halted. In recent years, with a deeper understanding of antibodies and immune cells, immunotherapy has emerged as a significant area of interest (*Reticker-Flynn and Engleman, 2020*). Despite this, the primary focus of research has been on T cells, with B cells receiving less attention. B cells play a crucial role in tumor development and treatment. Upon encountering antigens such as mutated proteins, B cells secrete antibodies (*Inoue and Kurosaki, 2024*). Tumorigenesis is a complex and dynamic process. As tumor cells start to develop, the structure of certain proteins changes due to mutations within these cells. These altered proteins can be recognized as non-self-antigens. However, some of these cells gradually adapt and manage to evade the body's immune system (*Zhu et al., 2021*). When the number of tumor cells surpasses a certain threshold, a tumor starts to form. Throughout this process, proteins within the tumor cells continuously accumulate various mutations to adapt to the immune system and the microenvironment (*Bozic et al., 2010*). Different epitopes on the mutated protein are exposed on the surface of tumor cells at various stages of the tumor. Occasionally, there is random exposure of these epitopes. This variability in epitope exposure is the primary reason for the immune system's inability to target the tumor effectively, leading to tumor immune escape and the failure of targeted drug treatments. In this study, we propose that these epitopes on the mutated proteins can be recognized as foreign objects, triggering an immune response and generating antibodies in a healthy individual. This leads to an intriguing question: Can B cells be leveraged in a healthy individual to generate anti-tumor antibodies from tumor-exposed individuals for cancer treatment? The proposed treatment mechanism involves these diverse antibodies binding to the corresponding epitopes of mutated proteins on the cancer cells. This binding could block cancer cell growth or activate signaling pathways, leading to cell death or apoptosis through antibody-dependent cell cytotoxicity. Moreover, we posit that solely living organisms have the capacity to generate a vast array and diversity of both known and unknown anti-tumor antibodies targeting these epitopes on the mutated proteins. The findings of this study substantiate the hypothesis. Currently, humans have the capability to synthesize certain known neoantigens to help the immune system launch the strongest attack against the tumor, such as the Moderna's cancer vaccine mRNA-415, which consists of a single synthetic mRNA coding for up to 34 neoantigens (*Liu and Ma, 2024*). However, as previously discussed, these artificially created antigens may lead to a situation similar to targeted therapy, where some tumor cells evade immune elimination.

This study has several limitations as follows: (1) The anti-tumor antibodies need to be identified. However, current methods for identifying both known and unknown antibodies pose a significant challenge. (2) Investigating immune response factors in the serum, such as cytokines, is crucial. However, there is a concern that the overall therapeutic effect may be compromised if antibodies and cytokines are considered separately. This concern stems from numerous studies, including those in traditional drug research, which have encountered failures when such a separation was attempted. The fundamental reason for this is that antibodies and cytokines form a mutually activating network. (3) Whole blood therapy may prove to be more effective due to the presence of immune cells such as T cells, B cells, and NK cells. These antibodies, cytokines, and immune cells form an interactive network that collaboratively works towards tumor reduction. However, while studying these components individually, it is crucial to consider the overall therapeutic effect. Furthermore, antibodies and cytokines that are either unknown or present in low concentrations should not be overlooked. (4) The impact of tumor cells on healthy mice is a critical factor to consider. The introduction of exogenous cells into the bodies of healthy mice may result in unpredictable outcomes (*Wei et al., 2021*). Two mice succumbed with weight loss, necessitating further investigation into the causes of this occurrence. Furthermore, is it possible for exogenous tumor cells to trigger immune storms or induce tumor formation in recipient mice? (5) In this study, only the blood type differences (type A or type B) of mice were considered, without taking into account other factors such as histocompatibility (*Yamamoto et al., 2001*). This is why paired mice were used in this study to reduce side effects. However, employing a completely random process for allocating the treatment groups would be preferable since it further underscores the efficacy and universality of serum therapy. (6) Although this treatment method has proven successful in mice, additional experiments are necessary before it can be applied to humans. For instance, the current ethical guidelines prohibit the injection of exogenous cells into the human body for the production of therapeutic serum. The complexity of the human body far exceeds that of mice, making it crucial to determine the appropriate dosage of tumor cells, the quantity of anti-tumor antibodies produced, and whether shortening or extending the duration of cell expansion (currently 7 days) or serum-based antibody production (also 7 days) would be more effective. (7) The question arises whether it would be beneficial to use serum treatments with antibodies derived from different animals. While this approach could potentially enhance treatment outcomes, it also introduces new challenges such as the selection of suitable animals, issues related to xeno-transplantation, and managing cross-species immune responses.

In conclusion, this research has explored a new strategy for mCSCC treatment by generating serum-based antibodies from tumor-exposed mice. The method involved the isolation of mCSCC cells, which were subsequently injected into healthy mice. This process stimulated the production of various anti-tumor antibodies present in the serum. These serums were then reintroduced into the tumor-bearing mice, effectively reducing the tumor volume. This cancer treatment method is very effective in treating mCSCC. However, certain aspects of the experiment warrant further investigation and resolution.

## Materials and methods
### DMBA/TPA carcinogenesis

Fifty C57BL/6 male mice were randomly divided into five groups: tumor + serum treatment, tumor + no serum treatment, control + serum treatment (control 1), control + no serum treatment (control 2), and serum provider. Each mouse from the tumor + serum treatment group was paired with a mouse of the same blood type (type A or type B) from the serum provider group. The mice in the tumor + serum treatment and tumor + no serum treatment groups received treatment with 7,12-Dimethylbenz(a) anthracene (DMBA) and 12-O-Tetradecanoylphorbol-13-acetate (TPA). The dorsal skin area of the mice was shaved. Two days later, the mice were topically treated with 60 μg of DMBA, dissolved in 200 μl of acetone, on their bare backs. This DMBA administration was carried out for 2 weeks, after which the mice were exposed to 2.5 μg of TPA in 200 μl of acetone once a week for a total of 10 weeks. DMBA (Lot: D3254) and TPA (Lot: P1585) were purchased from Sigma-Aldrich, China. Skin tumors were measured using a precision caliper, which allowed for the detection of size changes greater than 0.1 mm. Body weights were recorded weekly. Tumor volumes were measured on the first day of treatment and every week thereafter until the end of the experiments. The volume was

calculated using the formula V=π × [d² ×D]/6, where V represents the volume of the tumor, d is the minor axis of the tumor (the shortest diameter), D is the major axis of the tumor (the longest diameter) (*Lapouge et al., 2012*). *Figure 1* presents a workflow of this study. This study was performed in strict accordance with the recommendations in the Guide for the Care and Use of Laboratory Animals of the National Institutes of Health. All of the animals were handled according to approved institutional animal care and use committee protocols of Guilin Medical University. The protocol was approved by the Experimental Animal Ethics Committee of Guilin Medical University (Permit Number: GLMC202203177). All surgery was performed under sodium pentobarbital anesthesia, and every effort was made to minimize suffering.

## Cell preparation and serum injection

The preparation of single-cell suspensions from skin tumor tissues involved the use of a cell suspension preparation kit (Lot: KFS439, Beijing Baiaolaibo Technology Co, China), with a slight modification. Briefly, the dorsal skin tumor tissues were washed with PBS and cut into small fragments of 1–2 mm in size in a Petri dish containing EDTA/Trypsin. The minced tumor pieces were then transferred to a tube containing trypsin and incubated at 37 °C for an hour with shaking. DMEM/10% FBS was added to the dish to recover all cells and tissue, which were then passed through a 100 mm cell strainer. The cell suspension was centrifuged at 500×g for 5 min, and the recovered cells were plated out, ideally at densities of $1×10^5$ per 100 mm dish in KC growth medium (*Figure 2B*). The cells were then incubated at 37 °C in a 5% $CO_2$ incubator for 7 days with daily medium changes (*Li et al., 2017*). Each mouse in the tumor + serum treatment group was randomly paired with a mouse in the serum provider group. Approximately $5×10^5$ primary tumor cells suspended in PBS were injected into the tail vein of the paired mice in the serum provider group. After 7 days, 0.1 ml of whole blood was collected from the tail vein of the mice in the serum provider group under ether anesthesia. The serum was immediately separated by brief centrifugation, yielding about 0.02–0.05 ml of serum each time. This serum (0.02 ml) was then injected into the tail vein of its paired mouse in the tumor + serum treatment group once a week, for a total of three times (from weeks 15–17).

## Enzyme linked immunosorbent assay

Previous research has established a connection between the levels of p53, Bcl-xL, NF-κB, and Bax and the occurrence, progression, and metastasis of mCSCC (*Piipponen et al., 2021*; *Vasiljević et al., 2009*; *Han et al., 2019*; *Zhou et al., 2017*). Consequently, this study measured the concentrations of p53, Bcl-xL, NF-κB, and Bax in tissue samples using an ELISA assay. The ELISA Development Kits for mouse p53 (Lot: ab224878), Bcl-xL (Lot: ab227899), NF-κB (Lot: ab176648), and Bax (Lot: ab233624) were procured from Abcam, China. The procedure was as follows: The coated antibody was diluted and added to the ELISA plate (100 μL/well) and incubated for 48 hr at 4 °C. The ELISA plate was then washed three times with tris-buffered saline (TBS) and the diluted sample (100 μL/well) was added and incubated for 90 min at 37 °C. After washing three times, all samples were incubated with the diluted enzyme-labeled antibody (100 μl/well) for 60 min at 37 °C. The plate was washed three times again, and then the avidin-biotin-peroxidase complex (ABC) developer (100 μL/well) was added. After a 30 min incubation in the dark at 37°C, the reaction was stopped using 100 μl of stop buffer. Finally, the plates were read at 450 nm on a microplate reader (Thermo, China).

## Statistical analysis

Data are presented as mean ± standard deviation (SD) from three independent experiments. Differences before and after treatment were analyzed using paired sample *t*-tests with the SPSS 16.0 software package (SPSS Inc, Chicago). p value less than 0.05 was considered statistically significant. All experiments were repeated at least three times.

# Acknowledgements

This research was supported by the grants from National Natural Science Foundation of China (No.32260175), Guangxi Natural Science Foundation (No. 2018 GXNSFAA281048), and Guangxi Science and Technology Base and Special Fund for Talents (No. AD19110161).

# Additional information

### Funding

| Funder | Grant reference number | Author |
|---|---|---|
| National Natural Science Foundation of China | No.32260175 | Zheng Liu |
| Natural Science Foundation of Guangxi Zhuang Autonomous Region | No.2018GXNSFAA281048 | Zheng Liu |
| Specific Research Project of Guangxi for Research Bases and Talents | No.AD19110161 | Zheng Liu |

The funders had no role in study design, data collection and interpretation, or the decision to submit the work for publication.

### Author contributions

Zheng Liu, Conceptualization, Resources, Data curation, Supervision, Funding acquisition, Validation, Investigation, Methodology, Writing – original draft, Project administration, Writing – review and editing

### Author ORCIDs

Zheng Liu ⓘ https://orcid.org/0000-0003-4158-6768

### Ethics

This study was performed in strict accordance with the recommendations in the Guide for the Care and Use of Laboratory Animals of the National Institutes of Health. All of the animals were handled according to approved institutional animal care and use committee protocols of Guilin Medical University. The protocol was approved by the Experimental Animal Ethics Committee of Guilin Medical University (Permit Number: GLMC202203177). All surgery was performed under sodium pentobarbital anesthesia, and every effort was made to minimize suffering.

Combined public reviews: https://doi.org/10.7554/eLife.95678.3.sa1
Author response https://doi.org/10.7554/eLife.95678.3.sa2

# Additional files

### Supplementary files

• MDAR checklist

### Data availability

All data generated or analysed during this study are included in the manuscript and supporting files. Source data files have been provided for Figures 2 and 3.

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
