## [Editor Report · eLife assessment]

This study provides a **valuable** strategy for treating mouse cutaneous squamous cell carcinoma (mCSCC) with serum derived from mCSCC-exposed mice. The exploration of serum-derived antibodies as a potential therapy for curing cancer is particularly promising but the study provides **incomplete** evidence for specific effects of mCSCC-binding serum antibodies. This study will be of interest to scientists seeking a novel immunotherapeutic strategy in cancer therapy.

---

## [Referee Report · Combined public reviews]

Summary:

This study presents an immunotherapeutic strategy for treating mouse cutaneous squamous cell carcinoma (mCSCC) using a passive immunity-like strategy. The researcher induced tumors in healthy mice skin, then isolated the tumor cells and injected into other healthy mice to produce anti-tumor antibodies, and then administered these antibodies back into tumor-bearing mice. Results showed a reduction in tumor volume and altered expression of several cancer markers (p53, Bcl-xL, NF-κB, Bax). The analysis of results suggests a promising impact of antibody-rich serum in treating mouse cutaneous squamous cell carcinoma (mCSCC).

Strengths:

The approach does seem to have effect on preventing tumor progression, from both the tumor size and the cancer hallmarks expression level.

Weaknesses:

Despite the strength of the study, there are a few drawbacks in the study design and statistical analysis:

(1) Regarding the statistical analysis, the use of a paired t-test might be suboptimal for assessing the trend from weeks 15 to 17. It is recommended to consider alternative methods such as repeated measures ANOVA or linear regression to better capture and interpret the trend over this time period.

(2) To affirm the antibodies' role in the observed immune response, isolating antibodies rather than employing whole serum could provide more conclusive evidence. Comparative analyses with antibody-free serum or serum from healthy, non-immunized mice would clarify antibodies' specific contributions versus other serum components. The control group does not account for the potential immunostimulatory effects of serum injection itself. A better control would be tumor-bearing mice receiving serum from healthy non-mCSCC-exposed mice.

Response to author's rebuttal:

I acknowledge the value of evaluating serum therapy as a whole, considering the complex interactive networks and potential synergies involved. However, to scientifically understand and assess serum therapy, it remains essential to decompose the serum and identify the effective components. This decomposition would allow for a comparison of individual components with the overall effectiveness, thereby elucidating any synergistic effects.

While I agree that identifying specific epitopes and paratopes is indeed challenging and may exceed the scope of academic research, the use of methods such as Protein A purification or other techniques to isolate antibodies and cytokines from the serum is both necessary and feasible. This approach would enable a more detailed analysis of the individual effects of these components. I understand that the authors might not have that much resource, and I acknowledge this limitation. Nonetheless, other than this aspect, I believe the authors have adequately addressed my other concerns.

---

## [Author Response]

The following is the authors’ response to the original reviews.

**eLife assessment**
This study provides a useful strategy for treating mouse cutaneous squamous cell carcinoma (mCSCC) with serum derived from mCSCC-exposed mice. The exploration of serum-derived antibodies as a potential therapy for curing cancer is particularly promising but the study provides inadequate evidence for specific effects of mCSCC-binding serum antibodies. This study will be of interest to scientists seeking a novel immunotherapic strategy in cancer therapy.
**Joint Public Review:**
Summary:This study presents an immunotherapeutic strategy for treating mouse cutaneous squamous cell carcinoma (mCSCC) using serum from mice inoculated with mCSCC. The author hypothesizes that antibodies in the generated serum could aid the immune system in tumor volume reduction. The study results showed a reduction in tumor volume and altered expression of several cancer markers (p53, Bcl-xL, NF-κB, Bax) suggesting the potential effectiveness of this approach.Strengths:The approach shows potential effect on preventing tumor progression, from both the tumor size and the cancer biomarker expression levels bringing attention to the potential role of antibodies and B cell responses in cancer therapy.

We greatly appreciate your positive feedback on our study.

Weaknesses:These are some of the specific things that the author could consider to strengthen the evidence supporting the claims in their study.(1) The study fails to provide evidence of the specific effect of mCSCC-antibodies on mCSCC. The study utilized serum which also contains many immune response factors like cytokines that could contribute to tumor reduction. There is no information on serum centrifugation conditions, which makes it unclear whether immune components like antigen-specific T cells, activated NK cells, or other immune cells were removed from the serum. The study does not provide evidence of neutralizing antibodies through isolation, analysis of B cell responses, or efficacy testing against specific cancer epitopes. To affirm the specific antibodies' role in the observed immune response, isolating antibodies rather than employing whole serum could provide more conclusive evidence. Purifying the serum to isolate mCSCC-binding antibodies, such as through protein A purification, and ELISA would have been more useful to quantify the immune response. It would be interesting to investigate the types of epitopes targeted following direct tumor cell injection. A more thorough characterization of the antibodies, including B cell isolation and/or hybridoma techniques, would strengthen the claim.

I am deeply appreciative of the reviewer's highly professional comments. Tumor development involves the coexistence of cancer cells at different developmental stages, each harboring a variety of known and unknown mutated proteins. These mutated proteins expose multiple known and unknown epitopes, each capable of stimulating the production of corresponding antibodies in healthy mice. Identifying all these antibodies presents a significant challenge. Current research methodologies, such as ELISA, WB, and ChIP, can only identify known antibodies based on existing antigens. A prerequisite for using these techniques is that both antigens and antibodies are identified. At present, there is no technology available to identify antibodies produced by an unknown mutated protein and epitope. However, I find the reviewer's comments insightful. Perhaps we can initially identify some known mCSCC-antibodies on mCSCC. However, studying the specific effect of these known mCSCC-antibodies on mCSCC is uncertain because we believe that tumor shrinkage results from the combined action of both known and unknown antibodies.

We concur with the reviewer's observations regarding the use of serum, which is rich in immune response factors such as cytokines that could potentially contribute to tumor reduction. In our future research, we plan to systematically analyze the individual roles of these antibodies and cytokines in tumor reduction. In 1973, Nature published a report indicating that serum demonstrated promising results in tumor treatment (Immunotherapy of Cancer with Antibody in Rats. Nature 243, 492 (1973). https://doi.org/10.1038/243492b0). Since then, there have been scarcely any reports on serum therapy for tumors. The primary focus of our study is to evaluate the efficacy of serum therapy in treating tumors. We hypothesize that antibodies and cytokines form a complex interactive network, working in synergy to reduce tumors. Consequently, we believe that studying these antibodies and cytokines in isolation may not yield effective results.

In this study, the methodology section outlines the process of serum preparation. It is important to note that serum is devoid of blood cells. I hypothesized that whole blood might have superior therapeutic effects compared to serum. This is because antibodies could potentially synergize with immune cells (including T cells, B cells, and NK cells), thereby enhancing the effectiveness of the treatment. As previously discussed, these antibodies, cytokines, and immune cells form a complex interactive network aimed at tumor reduction. Consequently, there are numerous factors that could influence the experimental outcomes, which presents a challenge for analyzing the results. Furthermore, the implementation of whole blood transfusion therapy introduces additional considerations, such as potential side effects and reactions associated with blood transfusions.

We thank the reviewers for their suggestion to purify the serum in order to isolate mCSCC-binding antibodies. As we previously mentioned, separating a large number of both known and unknown serum antibodies presents a significant technical challenge. We are eager to discuss and consider suggestions from the reviewers regarding methods to identify a large variety and number of unknown antibodies on cells. Perhaps, as the reviewer suggested, we could begin with known antibodies and employ Protein A purification technology to purify these antibodies and subsequently detect immune responses. We could also categorize the types of epitopes targeted, direct tumor cell injection, to study the epitopes of these types in further studies. The suggestion to study the response of B cells is valuable, and we plan to conduct comprehensive research on the response and status of B cells in our future studies.

The purification of antibodies to enhance the specificity of their effectiveness against tumors is a critical aspect of our study. However, we would like to address some concerns raised. (1) The separation of all antibodies and cytokines presents a significant technical challenge. Particularly, there is a risk of overlooking antibodies that are present in low concentrations but play crucial roles. (2) What concerns us is that studying the composition separately would lose the overall effectiveness of the study. Our primary concern is that studying these components in isolation could compromise the holistic understanding of the study. This is akin to current research on traditional medicine, where the separation and individual study of compounds often result in a loss of overall therapeutic efficacy. For instance, consider a scenario where 100 antibodies collectively work to shrink a tumor. These antibodies interact with 20 cytokines, forming a complex network that enhances the cytokines' activity against tumor cells. Furthermore, many important antibodies and cytokines are currently unknown. Studying these antibodies in isolation could potentially result in the loss of this therapeutic effect. Therefore, in the discussion section, we have emphasized that our study considers a tumor mass, including tumor cells at various stages of development, as a single entity. As a practicing clinician, my primary focus is on the therapeutic outcomes in tumor treatments, despite the mechanisms of serum therapy remaining largely elusive, liking a black box.

(2) In the study design, the control group does not account for the potential immunostimulatory effects of serum injection itself. A better control would be tumor-bearing mice receiving serum from healthy non-mCSCC-exposed mice. Additionally, employing a completely random process for allocating the treatment groups would be preferable. Also, the study does not explain why intravenous injection of tumor cells would produce superior antibodies compared to those naturally generated in mCSCC-bearing mice.

I concur with the reviewer's perspective that using serum from healthy, non-mCSCC exposed mice as a control could potentially improve our study. Initially, our primary concern was to minimize harm to the mice and avoid excessive blood reactions, which led us to exclude the use of serum from healthy, non-mCSCC exposed mice in our control group. The main objective of our study was to investigate tumor shrinkage through serum treatment, specifically serum-derived antibodies. We anticipated that tumor-bearing mice receiving serum from healthy, non-mCSCC exposed mice would exhibit a response to the injected serum, which would manifest as a blood reaction. However, we did not expect this to result in a tumor treatment effect. If it turns out that normal serum (from healthy, non-mCSCC-exposed mice) possesses tumor-reducing properties, it would indeed be a novel discovery. We appreciate the reviewer's insightful suggestion and will consider incorporating it into our future research.

We concur with the reviewer's observations that the use of a completely random process for assigning treatment groups would be more desirable. Indeed, the complete randomization of the entire process further underscores the efficacy and universality of serum therapy. In this study, we utilized paired mice to mitigate the risk of cross-infection and adverse reactions associated with blood transfusions. We deeply value the reviewer's expert feedback.

Lastly, the reason why tumor cells, when intravenously injected, produce antibodies superior to those naturally generated in mCSCC-bearing mice, is due to the following reasons. As tumor cells grow, they produce a variety of mutated proteins to adapt to the immune microenvironment and evade the immune system of mCSCC-bearing mice. However, these tumor cells with mutated proteins are exceptionally sensitive and recognizable to healthy mice. This recognition triggers an immune response in healthy mice, leading to the production of specific therapeutic antibodies. This simultaneous production of diverse and abundant antibodies is only achievable by living organisms.

(3) In Figure 2B, it would be more helpful if the author could provide raw data/figures of the tumor than just the bar graph. Similarly in Figure 3, the author should show individual data points in addition to the error bar to visualize the actual distribution.

Raw data (numerical values) have been incorporated into Figures 2B and 3, but the data is placed in the table below the graph. If placed above the error bar, it requires a small font and may not be clear.

(4) The author mentioned that different stages of tumor cells have different surface biomarkers. Therefore, experimenting with injecting tumor cells at various stages could reveal the most immunogenic stage. Such an approach would allow for a comparative analysis of immune responses elicited by tumor cells at different stages of development.

Yes, throughout the course of tumor development, tumor cells at various stages will exhibit distinct markers or possess different mutated proteins. The concept of segregating tumor cells from different stages and independently comparing their immune responses is indeed commendable. Future research could involve isolating cells that express identical biomarkers at each stage for a comparative analysis of the immune responses triggered by the tumor cells. However, this approach diverges from the original intent of this study.

Most tumor cells exist within the same developmental stage. However, this does not imply that all tumor cells within the tumor mass are at the same stage. For instance, a stage III liver cancer tumor may contain both stage I and stage IV tumor cells. Moreover, due to the complexity of tumor development, not all tumor cell surface markers are identical, even for tumors at the same stage. For instance, 20 major proteins and 100 minor proteins are implicated in tumor formation. In fact, random mutations in just 5 of these major proteins and 10 minor proteins can instigate the development of tumors. This implies that the protein pattern (tumor cell surface markers) associated with each individual's tumor is unique. While studying tumor cells at different stages separately allows for the observation of the immune response of tumor cells at each stage, it lacks a comprehensive research and treatment effect. For this reason, the design of this study treats a tumor mass as a whole, encompassing both the primary stage tumor cells and those not in that stage. These tumor cells are then injected to produce corresponding therapeutic antibodies. Furthermore, if tumor cells from only one stage are isolated and specific antibodies are produced against these cells, it could lead to immune escape of tumor cells at other stages, preventing the tumor from shrinking. Therefore, our approach aims to address this issue by considering the tumor mass as a whole.

(5) In the abstract the author mentioned that using mCSCC is a proof-of-concept for this potential cancer treatment strategy. The discussion session should extend to how this strategy might apply to other cancer types beyond carcinoma.

We have incorporated an additional paragraph in the discussion section where we delve into the concepts and experimental principles underpinning this study. This, we believe, addresses the reviewer's query regarding the applicability of our study's methodology to other types of tumors. The process for other tumors also involves isolating cells from the tumor, stimulating therapeutic antibody production in healthy mice using these cells, and ultimately reintroducing these antibodies into mice with tumors to facilitate tumor elimination

**Recommendations For The Authors:**
The author is encouraged to refine the study's design in future studies considering the weaknesses highlighted above, summarize the results more effectively, and seek opportunities to expand on this promising idea and enhance the research's impact and applicability.

We greatly appreciate the valuable suggestions provided by the editor and reviewers. These insights will certainly be addressed in our future research endeavors.

Suggestions for title modification:Following the scope of the study, the term 'specific homologous neutralizing-antibodies' may be misleading as neutralizing antibodies typically refer to antibodies preventing viral cell entry. In cancer therapy, 'neutralization' is not a relevant concept, as cancer cells do not infect host cells. Using whole tumor cells as immunogens diverges from the specificity of traditional vaccination approaches that utilize well-defined proteins or antigens. Furthermore, the term "homologous" suggests a precision in targeting that is not demonstrated by reintroducing serum without isolating its specific components. Therapeutic effects should not be attributed to "neutralizing antibodies" without isolating or characterizing the antibody response or verifying their efficacy against specific cancer epitopes. Additionally, it is suggested that you indicate the biological system that your study utilised in the title. More so, this approach is not entirely novel, as seen with the use of adjuvants in some flu vaccines, or in Moderna's cancer vaccine mRNA-4157, which encodes up to 34 patient-specific tumor neoantigens. You can consider the title below or a variant of the same.Suggested title: Generating serum-based antibodies from tumor-exposed mice: a potential strategy in cutaneous squamous cell carcinoma treatment

I concur with your suggestion and have modified the title to " Generating serum-based antibodies from tumor-exposed mice: a new potential strategy for cutaneous squamous cell carcinoma treatment ". I believe this research remains some new, hence the addition of the word "new". Furthermore, the term "novel" in the paper has been either removed or substituted.

Moreover, I propose that this study shares similarities with Moderna's cancer vaccine mRNA-415, albeit with certain differences. Moderna's cancer vaccine mRNA-415 encodes 34 recognized neoantigens to stimulate an immune response by eliciting specific T cell responses. This is similar to the strategy of some companies developing a protein set for diagnosing lung cancer, liver cancer, among others. Without a doubt, these methods have improved the effectiveness of tumor diagnosis and treatment. However, I think that these methods currently face challenges in completely eradicating tumors because they perceive tumors as a static process and cells that express certain mutated proteins in a fixed manner. I believe that small molecule antibodies, cytokines, and immune cells present in serum that are difficult to detect, have low concentrations, or are unknown are essential for maintaining the expression of important mutant proteins and the escape of tumor cells. This is also the primary reason why tumors are difficult to treat and prone to recurrence at present.

From my perspective, different tumors, as well as different stages of the same tumor, express varying mutated proteins or surface markers. Targeting some may result in others escaping or even creating a more conducive growth environment for those that do escape. Our study adopts a comprehensive view of a tumor block, encompassing tumor cells at different stages and tumor cells at the same stage but expressing different biomarkers. This approach generates a multitude of known and unknown antibodies that work in concert with cytokines and immune cells. While our method may not be capable of generating all mutated proteins and epitope antibodies due to the weakness of some antigens (epitopes of mutated proteins), it can still be effective. As long as the number of tumor cells is reduced below a certain threshold following multiple rounds of treatment with various antibodies produced at different stages, these cancer cells can be eradicated by the body's immune system. This is a process that is real-time and dynamic. Undoubtedly, if it becomes evident that alterations in a set of proteins can bolster the immune system and eradicate tumor cells, then the implications are significant. The immunotherapy proteins, which have demonstrated positive therapeutic effects, developed by certain companies are also predicated on this very principle.

Finally, I greatly appreciate your suggestions, which will be considered and gradually addressed in future research.